# Breathing–Swallowing discoordination after definitive chemoradiotherapy for head and neck cancers is associated with aspiration pneumonia

Takuya Yoshida[1,2]*, Naomi Yagi[3], Takenori Ogawa[4]*, Ayako Nakanome[2], Akira Ohkoshi[2], Yukio Katori[2], Yoshitaka Oku[5]*

1 Department of Otolaryngology, Iwate Prefectural Iwai Hospital, Ichinoseki, Iwate, Japan, 2 Department of Otolaryngology-Head and Neck Surgery, Tohoku University Graduate School of Medicine, Sendai, Miyagi, Japan, 3 Advanced Medical Engineering Research Institute, University of Hyogo, Himeji, Hyogo, Japan, 4 Department of Otolaryngology, Gifu University Graduate School of Medicine, Gifu, Japan, 5 Department of Physiology, Hyogo College of Medicine, Nishinomiya, Hyogo, Japan

* taku.yoshi.1213@gmail.com (TY); yoku@hyo-med.ac.jp (YO); md395220@gmail.com (TO)

**Data Availability Statement:** Data cannot be shared publicly because they are derived from an individual patient. Data are available from the Ethics

## Abstract

### Purpose

Swallowing dysfunction and the risk of aspiration pneumonia are frequent clinical problems in the treatment of head and neck squamous cell carcinomas (HNSCCs). Breathing–swallowing coordination is an important factor in evaluating the risk of aspiration pneumonia. To investigate breathing–swallowing discoordination after chemoradiotherapy (CRT), we monitored respiration and swallowing activity before and after CRT in patients with HNSCCs.

### Methods

Non-invasive swallowing monitoring was prospectively performed in 25 patients with HNSCCs treated with CRT and grade 1 or lower radiation-induced dermatitis. Videoendoscopy, videofluoroscopy, Food Intake LEVEL Scale, and patient-reported swallowing difficulties were assessed.

### Results

Of the 25 patients selected for this study, four dropped out due to radiation-induced dermatitis. The remaining 21 patients were analyzed using a monitoring system before and after CRT. For each of the 21 patients, 405 swallows were analyzed. Swallowing latency and pause duration after the CRT were significantly extended compared to those before the CRT. In the analysis of each swallowing pattern, swallowing immediately followed by inspiration (SW-I pattern), reflecting breathing–swallowing discoordination, was observed more frequently after CRT (p = 0.0001). In 11 patients, the SW-I pattern was observed more frequently compared to that before the CRT (p = 0.00139). One patient developed aspiration pneumonia at 12 and 23 months after the CRT.

Committee of Tohoku University Hospital for researchers who meet the criteria for access to confidential data. The phone number is +81-22-717-8007 and the email address is med-kenkyo@grp.tohoku.ac.jp.

**Funding:** The author(s) received no specific funding for this work.

**Competing interests:** The authors have declared that no competing interests exist.

## Conclusion

The results of this preliminary study indicate that breathing–swallowing discoordination tends to increase after CRT and could be involved in aspiration pneumonia. This non-invasive method may be useful for screening swallowing dysfunction and its potential risks.

## Introduction

Head and neck squamous cell carcinoma (HNSCC) is a collective term for malignant neoplasms arising from the epithelia of the oral cavity, nasopharynx, oropharynx, hypopharynx, and larynx. Head and neck cancer accounts for 890,000 new cases and 450,000 deaths annually, representing 4.5% of all cancers, and ranking as the seventh most common cancer worldwide as of 2018 [1, 2]. Historically, smoking and alcohol exposure have been considered carcinogenic factors [3, 4]. Recent epidemiological trends have indicated a decline in the incidence of tumors in the oral cavity, hypopharynx, and larynx. However, oropharyngeal cancers alone are on the rise, and are attributed to human papillomavirus infection [5]. Radiotherapy (RT), particularly chemoradiotherapy (CRT), is a standard treatment option for head and neck squamous cell carcinoma (HNSCCs), and it plays an important role in organ preservation therapy, with up to 85% of patients diagnosed with HNSCCs receiving (C)RT as part of their treatment [6]. Chemoradiotherapy is essential for the treatment of head and neck cancer, not only in new cases but also in the postoperative period; this is because a radiation dose with cisplatin is superior to that of postoperative radiotherapy in terms of locoregional control and disease-free survival [7]. Although advances in treatment have improved survival rates, side effects are often chronic and persistent [8]. Typical side effects of CRT include mucositis, pain, xerostomia, edema, long-term muscle atrophy, fibrosis, and sensory loss, all of which decrease a patient's quality of life [9–14]. In addition to these changes, radiation-induced fibrosis of the pharyngeal mucosa occurs, resulting in decreased tongue strength, reduced tongue base retraction, delayed laryngeal vestibular closure, or problems with swallowing coordination movements, which are some of the most prominent symptoms [15–22]. Consequently, the risk of aspiration and aspiration pneumonia (defined as pneumonia secondary to the inhalation of food particles, saliva, or other foreign objects) increases. The problems associated with swallowing dysfunction are further compounded by the increased risk of aspiration pneumonia due to dysphagia [18, 23]. Reported frequencies of dysphagia after RT and CRT were as high as 52% and 69%, respectively. Additionally, aspiration pneumonia increases the risk of death by 42% in patients treated for head and neck cancer, and it accounts for 19% of all non-cancer deaths [24, 25]. Thus, the identification of dysphagia and prevention of aspiration pneumonia are of paramount importance to primary care providers.

The coordination of breathing and swallowing in the pharyngeal phases prior to the esophageal phase is a physiological defense mechanism that prevents aspiration and aspiration pneumonia. Swallowing usually occurs during expiration and subsequent breathing resumes with expiration, and this expiratory-swallow-expiratory (E-SW-E) pattern prevents entry of pharyngeal contents into the lower respiratory tract [26, 27]. The expiratory flow surrounding swallowing prevents the entry of pharyngeal contents into the lower airways. The expiratory flow also facilitates mechanical functions favorable to swallowing, such as elevation and closure of the larynx, generation of pharyngeal pressure with resultant food mass clearance, and opening of the pharyngo-esophageal segment [28]. In previous studies, it has been reported that patients with dysphagia following radiotherapy for head and neck cancer exhibited decreased

cough strength and expiratory force, highlighting the significance of expiration as a crucial defensive factor in swallowing [29]. Conversely, increased incongruence between breathing and swallowing, detected as I-SW (inspiration-swallow) and SW-I (swallow-inspiration) patterns, is a major risk factor for aspiration pneumonia in patients [26, 27, 30]. Technological advances over the past decade have led to the development of a variety of new devices in the head and neck region especially in surgical device, including the EXOSCOPE which allows the surgeon to work with high-definition images, to achieve a less invasive, more accurate, and safer approach for preserving function after treatment [31]. However, the gold standards for assessing swallowing function remain videoendoscopy (VE) and videofluoroscopy (VF) [32, 33], and the methods described above for assessing breathing and swallowing coordination are impractical. Recently, Yagi et al. developed a swallowing monitoring system as a non-invasive method to examine swallowing-respiration coordination [34]. This system has made it possible to systematically evaluate swallowing sounds, laryngeal movements during swallowing, and coordination of swallowing and breathing at the bedside. Subsequent studies using this system revealed that patients with dysphagia tend to have prolonged swallowing latency and pause duration, and exhibit I-SW or SW-I patterns that reflect dyscoordination of breathing and swallowing [30]. Furthermore, previous studies have shown that breathing-swallowing discoordination is a strong independent predictor of exacerbations in patients with chronic obstructive pulmonary disease (COPD), suggesting that breathing-swallowing discoordination warrants early detection and intervention [35]. However, there is insufficient information regarding the discordance between swallowing and breathing after CRT in patients with HNSCC who often experience dysphagia and subsequent aspiration pneumonia. In the present study, we used a non-invasive swallowing monitoring system to investigate breathing and swallowing discoordination related to CRT, and to monitor breathing and swallowing activity before and after CRT in patients with HNSCC.

## Methods and materials

### Patients and methods

This prospective, single-center, observational study enrolled patients with laryngeal, oropharyngeal, or hypopharyngeal squamous cell carcinoma who underwent definitive CRT between March 2017 and September 2018 at the Tohoku University Hospital. The recruitment period for this study was between December 2016 and December 2018. Patients with previous medical history or current diagnosis of dysphagia, respiratory disease, multiple concurrent primary cancers, chronic heart failure, uncontrolled infections, or autoimmune diseases were excluded from the study due to the potential augmentation of the risk for baseline breathing–swallowing discoordination. Chemoradiotherapy consisted of 70 Gy RT and cisplatin (100 mg/m$^2$) every three weeks. Swallowing was assessed at two time points: pre-treatment and one month after the CRT. The study protocol was approved by the local ethics committee of the Tohoku University of Medicine (#2016-1-578) on December 12, 2016. Written informed consent was obtained from all the patients.

### Swallowing evaluation

All swallowing evaluation were performed on the same day before and one month after CRT. The means of the assessment and outcomes were defined as follows:

(1) Ingestion status: Food Intake LEVEL Scale (FILS) [36]

This scale assesses the severity of dysphagia by evaluating the extent to which the patients consume food daily. Levels 1–3 pertain to diverse degrees of nonoral feeding. Levels 4–6 correspond to diverse degrees of oral food intake and alternative nutrition, respectively. Levels 7–9

correspond to diverse degrees of exclusive oral food intake, with Level 10 denoting normal oral food intake. The food intake level scale rating was obtained in via clinical interview and chart review.

(2) Swallowing dysfunction

Three experienced Otolaryngology specialists, who have completed swallowing function evaluation training set by the Ministry of Health, Labor, and Welfare and the Society of Swallowing and Dysphagia of Japan, along with a dysphagia-certified nurse and a speech-language pathologist, assessed the swallowing function using VE and VF. All evaluations of VE and VF were recorded in audio-video interleave (AVI) files at a rate of 30 frames per second. All evaluations were discussed on the same day following the examination to achieve inter-rater agreement. VE and VF were performed safely on all patients with no apparent complications.

For the VE evaluation, we used a nasopharyngeal-laryngoscope with a diameter of 3.9 mm with up/down tip deflection capability (Olympus ENF-VH; Olympus Tokyo, Japan) and a digital color video monitor to perform the VE. Patients in the sitting position underwent transnasal endoscopic examinations without nasal anesthetic spray. The Hyodo score was obtained based on endoscopic evaluation of swallowing using 3 ml blue-dyed water with direct visualization of the larynx. This method comprises four parameters: (1) the accumulation of saliva in the vallecula and piriform sinuses, (2) the induction of the glottal closure reflex by stimulating the epiglottis or arytenoid with an endoscope, (3) the initiation of the swallowing reflex measured by the timing of "white-out" (defined as the period during which the endoscopic image is obscured due to pharyngeal closure), and (4) the clearance of the pharynx after swallowing blue-dyed water. Each parameter is evaluated on a 4-point scale (0; normal, 1; mildly impaired, 2; moderately impaired, 3; severely impaired). The Hyodo score is the sum of scores for these parameters, ranging from 0 to 12. Patients with a score below 5 are considered to have normal swallowing function and can consume food orally without restrictions. Patients with a score above 8 are deemed to have severe swallowing dysfunction and are not permitted any oral intake [37].

Penetration-aspiration scale (PAS): PAS is an eight-point scale used to characterize both the location of airway invasion events and a patient's response with 1 representing the least and 8 representing the highest or most severe score. PAS scores encompass several observations within each score assessed by VF: (1) the depth of airway invasion (material positioned above, in contact with, or below the level of the vocal folds); (2) the presence or absence of material remaining after the swallow (ejected or not ejected); and (3) the patient's response to material in the airway (efforts to clear the material) [38]. For the VF evaluation, 3 cm3 of 40% (w/v) barium sulfate (Kaigen Pharma Co., Ltd, Osaka, Japan) with a viscosity of 16 mPa・s was injected into the participant's oral cavity while the evaluator was seated. Then, patients were observed during swallowing to determine whether there was penetration or aspiration, rated by the PAS. The visualization field of fluoroscopic examination extended from the infra-orbital border to the thoracic esophagus, and evaluations were conducted twice each for anterior and lateral views.

(3) The 10-item Eating Assessment Tool (EAT-10) [39]

The EAT-10 questionnaire comprises 10 inquiries, as delineated below, with responses graded on a scale from 0 to 4, and the total score was evaluated in the statistical analysis: (1) My swallowing problem has caused me to lose weight; (2) My swallowing problem interferes with my ability to go out for meals; (3) Swallowing liquids takes extra effort; (4) Swallowing solids takes extra effort; (5) Swallowing pills takes extra effort; (6) Swallowing is painful; (7) The pleasure of eating is affected by my swallowing; (8) When I swallow food sticks in my throat; (9) I cough when I eat; and (10) Swallowing is stressful. The score for each item on the EAT10 is 0 indicating no abnormality and 4 indicating severe disability. The cumulative score

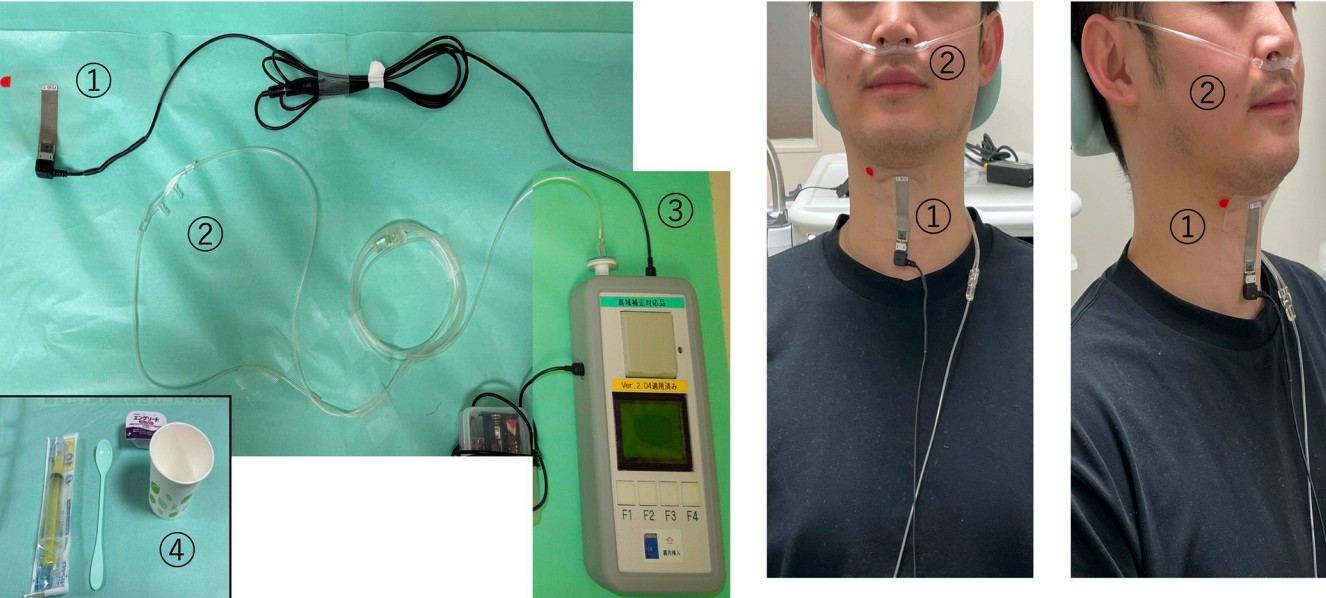

**Fig 1. Practical measurement of respiration-swallowing patterns.** 1) Probe to detect laryngeal elevation and swallowing motion; 2) Nasal cannula for airflow sensing; 3) Recording device; and 4) Other preparation items (test food, cups, and spoons).

ranges between 0 and 40. Elevated scores correlate with heightened severity of dysphagia, whereas diminished scores denote a milder manifestation of dysphagia. A score of $\geq 3$ is considered abnormal.

## Monitoring of swallowing

Three signal components were recorded by a swallowing monitoring system to detect and evaluate swallowing activity [34]. Fig 1 shows the analytical instrument that was used, and its attachment. Laryngeal motion and swallowing sounds were simultaneously recorded using a custom-made piezoelectric sensor attached to the thyroid cartilage. The sensor is a piezoelectric film (detector size: $10 \times 30$ mm) that generates an electric charge when bent, and has a wide dynamic range between 0–4 kHz. Respiratory flow was measured using a nasal cannula-type flow sensor (Pro-Tech ProFlow cannula, Sleep Lab Products, USA) and differential pressure transmitter (KL-17; Nagano Keiki, Japan), and was recorded at 1 kHz. Laryngeal motion was recorded at 1 kHz, and the sound signal was recorded at 10 kHz and stored simultaneously with the respiratory signal in a Micro SD card for later analysis. Additionally, we recorded the timing of swallowing for later verification using a foot switch to generate TTL-level pulse signals. The signals were analyzed using MATLAB (R2014b, MathWorks, USA) on a 64-bit Windows 8 professional computer. The stored data were subsequently analyzed by a medical engineering research expert and a neurophysiologist specialized in swallowing and respiration. Both analysts were blinded to patient information, and the results were determined through consensus.

The relationship between swallowing and respiratory cycles was assessed using parameters, including swallowing latency, laryngeal rising time (LRT), laryngeal activation duration (LAD), pause duration, old phase, and co-phase, as previously described [27, 30, 34, 40].

The parameters were defined as follows:

Swallowing latency: The time from the onset of respiratory pause to the onset of the swallowing reflex, defined as the time point when the laryngeal elevation speed reaches its maximum.

LRT: The time required for the larynx to elevate to its highest position.

LAD: The duration between the time of maximal laryngeal elevation speed and return of the larynx to its initial position.

Pause duration: The duration of respiratory pause associated with swallowing.

Old phase: Timing of swallowing in the respiratory cycle, expressed as the onset of the preceding inspiration to the maximum speed of laryngeal elevation (swallowing onset), normalized by the mean length of the respiratory cycle being 1.

Co-phase: The time from the onset of swallowing to immediately following inspiration, normalized by the mean length of the respiratory cycle being 1.

Breathing–swallowing (B-SW) coordination pattern: We classified the breathing–swallowing pattern based on two sets of parameters: (1) B-SW type, which characterizes the interplay between swallowing and the preceding respiratory phase, either E-SW (expiration-swallow) or I-SW (inspiration-swallow); and (2) SW-B type, which characterizes the relationship between swallowing and subsequent respiratory phase, either SW-E (swallow-expiration) or SW-I (swallow-inspiration).

Monitoring of swallowing was performed by two experienced otolaryngology specialists on the same day as the VE and VF evaluations. Before the examination, participants were told that the test would be performed with level 0 food, followed by water, that they would be asked to swallow five times each, that they could stop if they felt pain, that they should not talk during the test, and that they should not chew level 0 food. We utilized soft jelly called ENGE-LEAD®-grape(Otsuka Pharmaceutical Factory Inc.) and water as test foods, ensuring that the soft jelly properties, such as hardness, adhesiveness, and cohesiveness, strictly adhered to the criteria outlined in the Japanese Society of Dysphagia Rehabilitation specification for level 0 dysphagia diet in the International Dysphagia Diet Standardization Initiative report [41]. Participants were positioned upright in a chair, and they voluntarily swallowed approximately 3 g of level 0 test food from a teaspoon and 3 mL of water from a 5-mL syringe, repeating the process two to five times each. The reason for some variation in the protocol for administering the study diet is the establishment of criteria for halting the test in cases where clear signs of dysphagia are present and patient safety is deemed compromised. The participants were instructed to swallow the level 0 jelly without chewing. During the examination, the patients were not provided with specific verbal instructions regarding the nature or timing of their swallowing or breathing patterns relative to swallowing. Monitoring of swallowing was performed safely on all patients with no apparent complications.

## Statistical analyses

To compare variables before and after the CRT, t-tests were performed for swallowing latency, LRT, LAD, pause duration, old phase, and co-phase. Wilcoxon rank-sum tests were performed for the FILS, VE, VF, and EAT10. In the analysis of EAT-10, the total score was utilized. Comparisons of swallowing patterns (SW-E or SW-I) were performed using $\chi^2$ or Fisher's exact tests. The effect size (Cohen's d) was calculated to assess the substantive impact of CRT on each variable [42]. The categorization of effect sizes, as measured by Cohen's d, designates a lack of effect when <0.2, a small effect ranging between 0.2–0.5, a moderate effect spanning between 0.5–0.8, and a substantial effect at ≥0.8. The effect size for swallowing patterns was calculated using φ coefficient, which indicates correlation [43]. The categorization of effect sizes, as measured by φ coefficient, designates a lack of effect when <0.1, a small effect ranging between 0.1–0.3, a moderate effect spanning between 0.3–0.5, and a substantial effect at ≥0.5. All statistical analyses were performed using JMP11 software (SAS Institute Inc., Cary, NC, USA). In all instances, a two-sided p-value of <0.05 was considered statistically significant.

## Results

### Patient characteristics

Among the 25 patients selected for this study, four dropped out due to radiation-induced dermatitis. Thus, the analyses included 21 patients (19 men and 2 women) who underwent monitoring before and after CRT. All patients received CRT as initial treatment and no prior surgical intervention. The patients in this study were aged 35–73 years (median age, 62 years). Patient characteristics are shown in Table 1. Eleven patients had oropharyngeal cancer, four had hypopharyngeal cancer, and six had laryngeal cancer. In all the patients, the tumors were pathologically diagnosed as squamous cell carcinoma. The most common clinical stages were stages IVa and III. All the patients received CRT, with a high-dose cisplatin (CDDP) regimen (80–100 mg /m$^2$ body surface area of cisplatin infusion, three times weekly). One patient with bulky T2 hypopharyngeal cancer underwent CRT after a discussion with a multidisciplinary team. The median CDDP dose in all the 21 patients was 280 mg/m$^2$ (150–300 mg/m$^2$), and the total radiation dose was 70 Gy. One patient received a CDDP dose of 150 mg/m$^2$ for acute renal injury.

### Swallowing evaluation

The results of the FILS, VE, VF, and EAT10 are shown in Table 2. The FILS decreased, indicating that patients encountered some trouble with ingestion after CRT (p = 0.0009). Additionally, there was a significant increase in the Hyodo Score, suggesting impaired safe swallowing coordination as an objective finding on endoscopic examination after CRT (p = 0.0454). There was also a significant worsening observed in EAT10 after treatment, indicating a trend where patients themselves had trouble with swallowing after CRT (p = 0.0304). However, no significant changes were observed in the PAS (VF).

### Monitoring of swallowing

The results of the swallowing monitoring are presented in Table 3. Level 0 food, water, and both conditions combined showed significantly longer postoperative swallowing latency and pause duration than preoperative values (p<0.0001).

**Table 1. Patients' profiles.**

| Factors | Category | N or Median (range) |
|---|---|---|
| Gender | Male | 19 |
| | Female | 2 |
| Age (years) | | 62 (35–73) |
| Site | Oropharynx (p16 INK4a Negative) | 1 |
| | Oropharynx (p16 INK4a Positive) | 10 |
| | Hypopharynx | 4 |
| | Larynx | 6 |
| Clinical stage (UICC 7th Edition) | II | 1 |
| | III | 10 |
| | IVA | 10 |
| Total CDDP dose | 150 mg/m$^2$ | 1 |
| | 200–250 mg/m$^2$ | 9 |
| | 250–300 mg/m$^2$ | 11 |

Abbreviations: UICC, Union for International Cancer Control; CDDP, cisplatin

**Table 2. Evaluation of swallowing.**

| | Swallowing evaluation | | | |
|---|---|---|---|---|
| | Median (range) | | P value | Effect size (Cohen's d) |
| Parameters | Before CRT | After CRT | | |
| FILS | 10 (5–10) | 8 (3–10) | 0.0009 ** | 1.19 † |
| VE | 0.5 (0–3) | 1.5 (1–6) | 0.0454 * | 1 † |
| VF (PAS) | 0.5 (1–2) | 1.5 (1–6) | 0.206 | 0.77 † |
| EAT10 | 0.5 (0–12) | 9 (0–26) | 0.0304 * | 1.13 † |

Abbreviations: CRT, chemoradiotherapy, VE, videoendoscopy, VF, videofluoroscopy, FILS, Food Intake LEVEL Scale, EAT-10, 10-item eating assessment tool

*p < 0.05

**p < 0.01

†d>0.5.

Analysis of swallowing water and both conditions combined showed significantly prolonged old phase and co-phases after CRT. Each swallowing event was characterized by one of the following four patterns: I-SW, E-SW, SW-E, or SW-I, which indicate the order of swallowing (SW), inhalation (I), or exhalation (E), respectively. The I-SW pattern with level 0 food was observed in 5.56% and 9.64% of all swallowing events before and after the CRT, respectively. The I-SW pattern with water was observed in 15.05% and 20.79% of all swallowing events before and after the CRT, respectively. In the combined water and level 0 food analysis, the I-SW pattern was observed in 10.38% and 13.08% of all swallowing events before and after the CRT. Before the CRT, the SW-I patterns were observed in 1.01% and 0.52% of all swallowing events in water and all conditions, respectively, but not in the level 0 food condition. After the CRT, significantly more SW-I patterns were observed in level 0 food, water, and all conditions than before CRT, with statistical significance of p = 0.0025 (0% vs. 9.2%), p = 0.019 (1.01% vs. 8.57%), and p = 0.0001 (0.52% vs. 8.85%), respectively.

Among the 21 patients under consideration, 20 exhibited the SW-E pattern before the treatment, whereas only one exhibited the SW-I pattern. After the CRT, 10 individuals manifested the SW-E pattern, and 11 displayed the SW-I pattern. A pronounced disparity was observed after the treatment, with a notable increase in the prevalence of the SW-I pattern (p = 0.00139).

Fig 2 shows a representative E-SW-E pattern before the CRT, and Fig 3 shows a representative E-SW-I pattern after the CRT. One patient developed aspiration pneumonia at 12 and 23 months after the CRT. The diagnosis of aspiration pneumonia was made comprehensively, considering the patient's symptoms, clinical course including swallowing function leading to pneumonia, findings of pneumonia and its location on Computed Tomography imaging, and blood tests.

The expiration-swallow-expiration pattern is shown before CRT, and the expiration-swallow-inspiration pattern due to prolonged pause duration is shown after CRT. Irregular laryngeal sounds are associated with swallowing difficulty, resulting in a prolonged pause duration and an SW-I swallowing pattern.

LRT, laryngeal rising time; LAD, laryngeal activation duration; SW-I, inspiration occurs immediately after a swallow.

## Discussion

This observational study evaluated patients who underwent CRT to gain insight into the breathing–swallowing discoordination that occurs after CRT for HNSCCs.

**Table 3. Monitoring of swallowing.**

| | | | | | |
|---|---|---|---|---|---|
| Level 0 food | | | | | |
| | Mean(±SD) | | P value | Effect size (Cohen's d) | Effect size (φ coefficient) |
| Parameters | before CRT | after CRT | | | |
| swallow latency (ms) | 284.710 (±463.710) | 886.598 (±989.460) | < .0001** | 0.79 † | |
| LRT(ms) | 496.839 (±366.352) | 559.770 (±316.747) | 0.22 | 0.18 | |
| LAD(ms) | 895.752 (±342.183) | 852.908 (±321.319) | 0.388 | 0.13 | |
| pause duration (s) | 1.051 (± 0.666) | 1.853 (±1.340) | < .0001** | 0.76 † | |
| old phase | 0.881 (±0.447) | 1.042 (±0.697) | 0.066 | 0.33 | |
| co-phase | 0.683 (±0.314) | 0.751 (±0.520) | 0.288 | 0.16 | |
| E-SW (%) | 94.44 | 90.36 | 0.3913 | | 0.055 |
| I-SW (%) | 5.56 | 9.64 | | | |
| SW-E (%) | 100 | 90.8 | 0.0025** | | 0.196 ‡ |
| SW-I (%) | 0 | 9.2 | | | |
| Water | | | | | |
| | Mean(±SD) | | P value | Effect size (Cohen's d) | Effect size (φ coefficient) |
| Parameters | before CRT | after CRT | | | |
| swallow latency(ms) | 322.728 (±673.349) | 982.679 (±1393.792) | < .0001** | 0.6 | |
| LRT(ms) | 548.350 (±341.687) | 604.755 (±353.613) | 0.242 | 0.16 | |
| LAD(ms) | 923.612 (±339.923) | 901.132 (±324.905) | 0.625 | 0.07 | |
| pause duration(s) | 1.053 (±0.750) | 2.174 (±1.866) | < .0001** | 0.78† | |
| old phase | 0.722 (±0.450) | 0.916 (±0.577) | 0.0066** | 0.38 | |
| co-phase | 0.588 (±0.290) | 0.704 (±0.429) | 0.0225* | 0.32 | |
| E-SW (%) | 84.95 | 79.21 | 0.352 | | 0.061 |
| I-SW (%) | 15.05 | 20.79 | | | |
| SW-E (%) | 98.99 | 91.43 | 0.019* | | 0.152 ‡ |
| SW-I (%) | 1.01 | 8.57 | | | |
| All | | | | | |
| | Mean(±SD) | | P value | Effect size (Cohen's d) | Effect size (φ coefficient) |
| Parameters | before CRT | after CRT | | | |
| swallow latency(ms) | 304.688 (±582.089) | 939.368 (±1226.056) | < .0001** | 0.66 | |
| LRT(ms) | 523.908 (±353.633) | 584.477 (±337.380) | 0.0848 | 0.24 | |
| LAD(ms) | 910.393 (±340.408) | 879.394 (±323.349) | 0.358 | 0.09 | |
| pause duration(s) | 1.051 (±0.906) | 2.030 (±0.0913) | < .0001** | 0.77† | |
| old phase | 0.797 (±0.455) | 0.974 (±0.636) | 0.0017** | 0.32 | |
| co-phase | 0.633 (±0.304) | 0.726 (±0.472) | 0.0219* | 0.23 | |
| E-SW (%) | 89.62 | 84.24 | 0.163 | | 0.072 |
| I-SW (%) | 10.38 | 13.08 | | | |
| SW-E (%) | 98.48 | 91.15 | 0.0001** | | 0.185 ‡ |
| SW-I (%) | 0.52 | 8.85 | | | |

Abbreviations: CRT, chemoradiotherapy, LRT, laryngeal rising time, LAD, laryngeal activation duration

*p < 0.05

**p < 0.01

†d > 0.5

‡φ > 0.1.

Our results showed significantly prolonged swallowing latency and pause duration after CRT compared with those before CRT. This could potentially be attributed to reported adverse effects of CRT, such as decreased laryngeal perception, xerostomia, radiation-induced fibrosis

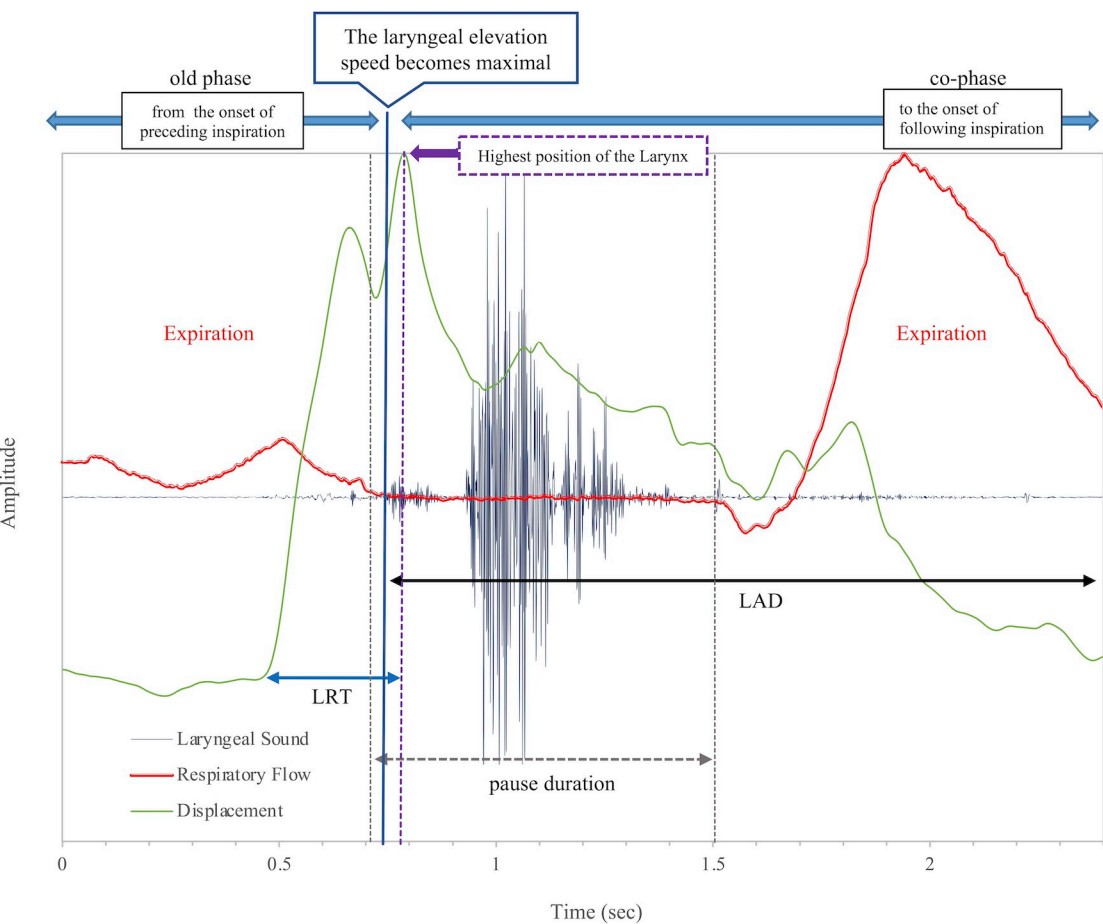

**Fig 2. Swallowing pattern in case #1 before CRT.**

of the mucosa associated with the pharyngeal wall, and decreased motility of the oropharynx [9, 10, 15, 16]. However, the exact mechanism was not elucidated in this study. Investigating the relationship between swallowing-related muscle strength, motility, laryngeal sensation, saliva volume, and breathing-swallowing coordination through simultaneous measurements would be a prospective focus for future research. Additionally, we observed a significant increase in the frequency of the SW-I patterns. One patient experienced repeated episodes of aspiration pneumonia during the clinical course. The increase in the SW-I pattern can be explained by prolonged swallow latency and pause duration [30, 44]. Although not a significant change, the I-SW pattern may be caused by an airway defense response that compensates for swallowing latency. In previous studies monitoring respiratory swallowing, the I-SW and SW-I patterns showed higher frequencies in patients with stroke (18.4%), Parkinson's disease (40.0%), and COPD (13.6%) [45–47]. These rates surpassed those observed in this study.

The prevalence of the SW-I pattern exhibited a notable increase following liquid ingestion in contrast to solid ingestion. This phenomenon could be attributed to differences in the physical attributes of materials, such as viscosity and shape stability, which influence swallowing dynamics, rendering liquids more challenging to swallow than solids [48]. Moreover, it has been suggested that level 0 food, which is deemed the safest for preventing aspiration, is particularly efficacious in risk assessment [35].

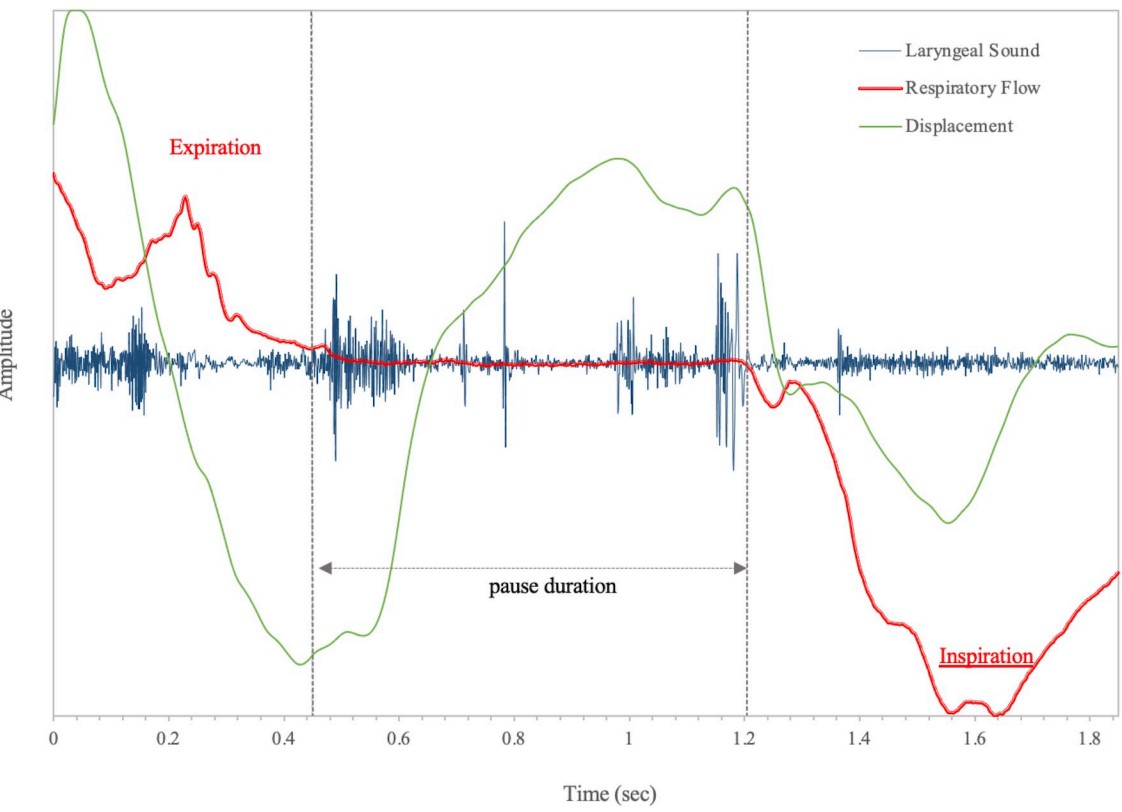

**Fig 3. Swallowing pattern in case #1 after CRT.**

Disruptions in the respiratory–swallow coordination pattern have been reported in patients with dysphagia following treatment for head and neck cancer, including surgical interventions [28]. A recent study also demonstrated prolonged swallowing latency, swallowing apnea, and increased post-swallowing inspiration in mouse models after CRT compared with control groups [49]. Although our investigation specifically focused on patients who underwent CRT as their initial treatment, the findings were aligned with the outcomes observed in previous studies.

Coordination between swallowing and breathing is regulated by the interaction of central pattern generators (CPGs) within the brainstem [50–53]. Swallowing can be regarded as an external stimulation and natural disturbance of the respiratory CPG. When a respiratory oscillator is affected by an external stimulus, the timing at which a specific phase begins is either advanced or delayed [11, 35, 40]. The prolongation of swallowing latency and pause duration affects the phase response characteristics, which are closely defined by respiratory CPGs, resulting in the SW-I pattern [30]. The variability in cortically controlled swallowing parameters, such as swallowing latency and pause duration, also affects phase response characteristics. In the present study, prolonged swallowing latency after CRT was attributed to edema, fibrous scarring, muscle weakness, and decreased laryngeal sensation due to chemoradiation treatment [12, 13]. Although the respiratory rhythm is centrally controlled, extrapulmonary peripheral changes associated with chemoradiation cause changes in swallowing timing and length; thus, inhalation after swallowing may occur due to phase resetting by the respiratory rhythm.

Hopkins-Rossabi et al. reported that in recent years, patients with HNSCCs with dysphagia experience increased occurrences of dysphagia and penetration/aspiration as the proportion of optimal respiratory-swallow phase patterns decreases. They also noted changes in respiratory-swallow phase patterns before and after cancer treatment. Our findings in this study are consistent with the results of this previous research [54]. Martin-Harris et al. demonstrated that a biofeedback approach for coordinating breathing and swallowing improved swallowing function in patients with postoperative patients with head and neck cancer [28]. Early detection and intervention of coordination disorders in breathing and swallowing may be valuable for improving the quality of life of patients. Based on our results, we believe that our methodology, which detects respiratory and swallowing dysfunctions, can serve as a biomarker for the risk of aspiration pneumonia after CRT.

To understand the neural mechanisms underlying the effects of CRT on respiratory–swallowing coordination, it is essential to simultaneously measure various parameters. This includes assessing the volume and electromyography of muscles related to swallowing, such as the tongue and pharynx, conducting objective evaluations of pharyngeal perception and saliva volume, and utilizing methods, such as esophageal manometry. This evaluation should be conducted comprehensively and longitudinally.

Moreover, the study participants should encompass a sufficient number of individuals with diverse backgrounds, including not only healthy individuals, but also patients with neurological disorders, COPD, head and neck cancer, and various other conditions.

Before and after CRT, deterioration in FILS and EAT10 scores was observed. This suggests that due to CRT, patients experience inhibition of oral intake, indicating the need for some form of nutritional support for the maintenance of vital functions. It also indicates a decrease in quality of life from both functional and psychological perspectives. A patient-completed swallowing screening questionnaire offers a simple and effective tool for daily clinical use, allowing healthcare providers to promptly identify swallowing issues post CRT. This could help in the timely implementation of further assessments such as VF and VE, along with appropriate rehabilitation interventions. In this protocol, the VE evaluation, or Hyodo score, significantly worsened before and after CRT, whereas no significant change was noted in the PAS scale. This suggests that when assessing swallowing after CRT, PAS may not detect changes or may be obscured, particularly with a small bolus condition. While swallowing evaluation with VE can be conducted at the bedside using only an endoscope, VF necessitates relocation to the fluoroscopy room and radiation exposure. In terms of ease of performance, VE and evaluation of breathing-swallowing coordination can be considered advantageous for evaluating swallowing function after CRT.

In this study, only small boluses were utilized. The rationale behind this decision was to ensure consistency with the protocols and conditions of prior studies employing noninvasive swallowing measurement system. This approach aimed to investigate variations in the coordination of breathing and swallowing across diverse patient conditions. Increasing bolus size is expected to prolong the time taken for bolus transit relative to smaller boluses, resulting in extended pause duration and potentially exacerbating in breathing-swallowing discoordination. Therefore, the observed breathing-swallowing discoordination in this study may be underestimated compared to evaluations using oral intake or protocols employing larger boluses in clinical practice. Although not included in this protocol, the utilization of various test foods in VE, VF and monitoring of swallowing could allow for the evaluation of swallowing function for each food type. Moreover, by utilizing these assessments to determine the viscosity of safe and suitable foods and rehabilitation strategies based on swallowing function, and subsequently enhancing the level of oral intake, it is conceivable that the quality of life for patients after CRT may be improved.

A limitation of this study is its short evaluation period. Thus, a long-term investigation of respiratory dysphagia and swallowing function after CRT is required. Further extensive studies are required to determine whether breathing and swallowing problems after CRT increase the risk of aspiration pneumonia. Based on the observational design of this study, there is a potential for selection bias, information bias, and involvement of latent confounding factors. Furthermore, a limitation arises from the absence of comparable control groups for comparison. Individual characteristics, including sex, age, primary site, and stage classification, exhibit variations, introducing potential limitations, such as selection and information biases. In this study, no assessment was conducted on indicators related to swallowing efficiency, such as swallowing residue. For patients at higher risk of aspirating post-swallow pharyngeal residue, such as in this study, it was deemed necessary to include the evaluation of swallowing residue as an additional assessment item for a more comprehensive evaluation. The measurements related to CRT such as pain and discomfort were not incorporated as evaluation criteria in accordance with the protocol. The possibility that these items could affect the results of this study cannot be ruled out. Finally, the study's statistical power and capacity to discern significant effects may have been compromised due to the relatively small sample size of 21 participants.

## Conclusions

These results suggest that monitoring swallowing is a useful method to detect changes in breathing–swallowing behavior before and after CRT, and may detect the risk of aspiration pneumonia following treatment.

## Supporting information

**S1 Checklist. Human participants research checklist.**
(DOCX)

## Acknowledgments

We would like to thank Editage (www.editage.jp) for English language editing.

## Author Contributions

**Conceptualization:** Takenori Ogawa, Yoshitaka Oku.

**Data curation:** Takuya Yoshida, Naomi Yagi.

**Formal analysis:** Takuya Yoshida, Naomi Yagi, Akira Ohkoshi.

**Project administration:** Takuya Yoshida.

**Resources:** Ayako Nakanome.

**Supervision:** Yukio Katori, Yoshitaka Oku.

**Visualization:** Takuya Yoshida.

**Writing – original draft:** Takuya Yoshida.

**Writing – review & editing:** Takuya Yoshida.

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
