## [Decision Letter · Decision Letter 0]

16 Nov 2023

PONE-D-23-27897Breathing-swallowing discoordination after definitive chemoradiotherapy for head and neck cancers is associated with aspiration pneumoniaPLOS ONE

Dear Dr. Yoshida,

Thank you for submitting your manuscript to PLOS ONE. After careful consideration, we feel that it has merit but does not fully meet PLOS ONE’s publication criteria as it currently stands. Therefore, we invite you to submit a revised version of the manuscript that addresses the points raised during the review process.

We look forward to receiving your revised manuscript.

Kind regards,

Antonino Maniaci

Academic Editor

PLOS ONE

Additional Editor Comments:

Please perform all the revisions required.

Reviewers' comments:

Reviewer's Responses to Questions

**Comments to the Author**

1. Is the manuscript technically sound, and do the data support the conclusions?

Reviewer #1: Yes

Reviewer #2: No

2. Has the statistical analysis been performed appropriately and rigorously? 

Reviewer #1: Yes

Reviewer #2: No

3. Have the authors made all data underlying the findings in their manuscript fully available?

Reviewer #1: Yes

Reviewer #2: Yes

4. Is the manuscript presented in an intelligible fashion and written in standard English?

Reviewer #1: Yes

Reviewer #2: Yes

5. Review Comments to the Author

Reviewer #1: The authors provided an interesting observational single center study on breathing-swallowing discoordination after definitive chemoradiotherapy for head and neck cancers. The manuscript provides original data on a novel topic. However, there are some issues that need to be addressed with a revision:

- The introduction section is quite brief. Please provide additional background in order to justify the rationale and aims of the study.

- Please provide the date of ethical committee approval.

- Please specify the institute were the study was conducted in the methods section.

- Plase use the same tamplate for all the tables.

- Table 3 should be revised as for layout.

- Table 4 should be unified as being a single table might increase readability.

- Please remove table 5 and provide its content in the text.

- The observational design of the study must be included among the limitations of the study.

Reviewer #2: Introduction

- Provide more background on the prevalence of dysphagia and aspiration pneumonia following chemoradiotherapy (CRT) for head and neck cancers. Cite key statistics on incidence rates. doi: 10.1002/hed.20279 and doi:10.1016/j.jvoice.2021.09.040

- Explain the pathophysiology leading to swallowing dysfunction after CRT - fibrosis, muscle weakness, sensory changes etc.

- Discuss the importance of breathing-swallowing coordination in airway protection and how disorders can increase aspiration risk.

- Introduce innovative techniques in H&N management like videofluoroscopy, endoscopy for swallow evaluation and esoscope to correct identificate digestive tract for correct reconstruction. Discuss and cite doi:10.3390/jcm11133639 and doi: 10.1007/s00455-022-10484-8.

- Discuss prior research that has analyzed breathing-swallowing patterns in patients with dysphagia using noninvasive monitoring methods.

Methods

- Explain swallowing evaluation methods like Food Intake Level Scale, Hyodo score, Penetration-Aspiration Scale in more detail.

- Provide more details on the swallow monitoring system - sensors used, data acquisition, analysis software etc.

- Clearly define all swallowing parameters that were quantified - latency, laryngeal motion, respiratory phases etc.

- Describe the food and liquid stimuli used for swallow trials along with preparation protocols.

- Elaborate on participant instructions prior to swallow trials and precautions taken.

- Explain the statistical tests used for comparing swallowing metrics before and after CRT.

Results

- When reporting changes in swallowing metrics before and after CRT, include exact p-values and effect sizes for the comparisons.

- For swallowing coordination patterns, provide percentages in addition to p-values when comparing groups.

- Consider including a table comparing swallowing parameters and coordination patterns before and after CRT.

- Mention any differences between liquid and solid swallows in the effects of CRT on breathing-swallowing coordination.

Discussion

- Interpret the prolonged swallowing latency and pause duration after CRT in context of known pathophysiological effects of chemoradiation on pharyngeal muscles and sensation.

- Compare the magnitude of increase in SW-I patterns to rates reported for other disorders like stroke, COPD.

- Discuss possible reasons for SW-I pattern increase specifically after liquid swallows versus solids.

- Compare your coordination findings to results from prior studies analyzing breathing-swallowing after chemoradiation.

- Discuss whether SW-I patterns could be a predictive biomarker of aspiration risk in this population.

- Suggest future studies to understand neural mechanisms underlying CRT effects on breathing-swallowing coordination.

6. PLOS authors have the option to publish the peer review history of their article (what does this mean?). If published, this will include your full peer review and any attached files.

Reviewer #1: No

Reviewer #2: No

---

## [Author Response · Author response to Decision Letter 0]

3 Jan 2024

Dear Editors and Reviewers

We sincerely appreciate the editor’s and reviewers’ comments. We revised our article according to the comments. 

Reviewer #1: 

The authors provided an interesting observational single center study on breathing-swallowing discoordination after definitive chemoradiotherapy for head and neck cancers. The manuscript provides original data on a novel topic. However, there are some issues that need to be addressed with a revision: 

Response: Thank you for your thoughtful review. 

Comment:

1. The introduction section is quite brief. Please provide additional background in order to justify the rationale and aims of the study.

Response: Thank you for your valuable opinion. The background was too short, so I have added citations and added more details.

2. Please provide the date of ethical committee approval.

Response: According to this comment, I added 2016-12-12.

3. Please specify the institute were the study was conducted in the methods section.

Response: According to this comment, I have written that the procedure was carried out at Tohoku University Hospital.

4. Plase use the same tamplate for all the tables.

Response: Thank you for your valuable comments. The table templates have been unified.

5. Table 3 should be revised as for layout.

6. Table 4 should be unified as being a single table might increase readability.

Response: I would like to thank you for your very valuable comments. Based on Reviewer 2's comments, we combined Tables 3 and 4 into one and adjusted the layout.

7. Please remove table 5 and provide its content in the text.

Response: According to this comment, Table 5 has been deleted and its contents have been written in lines 231 to 234 of the main text as shown below.

 “Among the 21 patients under consideration, 20 exhibited the SW-E pattern before the treatment, whereas only one exhibited the SW-I pattern. After the CRT, 10 individuals manifested the SW-E pattern, and 11 displayed the SW-I pattern. A pronounced disparity was observed after the treatment, with a notable increase in the prevalence of the SW-I pattern (p = 0.00139)”

8.　　The observational design of the study must be included among the limitations of the study.

Thank you for your valuable comments. We wrote it in lines 296 to 302 of the main text as shown below. 

“A limitation of this study is its short evaluation period. Thus, a long-term investigation of respiratory dysphagia and swallowing function after CRT is required. Further extensive studies are required to determine whether breathing and swallowing problems after CRT increase the risk of aspiration pneumonia. Individual characteristics, including sex, age, primary site, and stage classification, exhibit variations, introducing potential limitations, such as selection and information biases. Finally, the study’s statistical power and capacity to discern significant effects may have been compromised due to the relatively small sample size of 21 participants.“

Reviewer #2: 

Response: Thank you for your thoughtful review.

Comment:

Introduction

1. Provide more background on the prevalence of dysphagia and aspiration pneumonia following chemoradiotherapy (CRT) for head and neck cancers. Cite key statistics on incidence rates. doi: 10.1002/hed.20279 and doi:10.1016/j.jvoice.2021.09.040

Response: According to this comment, We have cited the papers you have presented (citation numbers 2 and 7) and have included the morbidity rate and basic information in the background n lines 52 to 55 of the main text.

2. Explain the pathophysiology leading to swallowing dysfunction after CRT - fibrosis, muscle weakness, sensory changes etc.

Response: According to this comment, we have provided detailed information on lines 65-73 of the main text, including cited references as shown below. 

“Typical side effects of CRT include mucositis, pain, xerostomia, edema, long-term muscle atrophy, fibrosis, and sensory loss, all of which decrease a patient’s quality of life [9-14]. In addition to these changes, radiation-induced fibrosis of the pharyngeal mucosa occurs, resulting in decreased tongue strength, reduced tongue base retraction, delayed laryngeal vestibular closure, or problems with swallowing coordination movements, which are some of the most prominent symptoms [15-22]. Consequently, the risk of aspiration and aspiration pneumonia (defined as pneumonia secondary to the inhalation of food particles, saliva, or other foreign objects) increases. The problems associated with swallowing dysfunction are further compounded by the increased risk of aspiration pneumonia due to dysphagia [18, 23].”

3. Discuss the importance of breathing-swallowing coordination in airway protection and how disorders can increase aspiration risk.

Response: According to this comment, we have provided detailed information on lines 78-86 of the main text, including cited references as shown below. 

“The coordination of breathing and swallowing is a physiological defense mechanism that prevents aspiration and aspiration pneumonia. Swallowing usually occurs during expiration and subsequent breathing resumes with expiration, and this expiratory-swallow-expiratory (E-SW-E) pattern prevents entry of pharyngeal contents into the lower respiratory tract [26, 27]. The expiratory flow surrounding swallowing prevents the entry of pharyngeal contents into the lower airways. The expiratory flow also facilitates mechanical functions favorable to swallowing, such as elevation and closure of the larynx, generation of pharyngeal pressure with resultant food mass clearance, and opening of the pharyngo-esophageal segment [28]. Conversely, increased incongruence between breathing and swallowing, detected as I-SW and SW-I patterns, is a major risk factor for aspiration pneumonia in patients [26, 27, 29].”

4. Introduce innovative techniques in H&N management like videofluoroscopy, endoscopy for swallow evaluation and exoscope to correct identificate digestive tract for correct reconstruction. Discuss and cite doi:10.3390/jcm11133639 and doi: 10.1007/s00455-022-10484-8.

Response: According to this comment, we have quoted the papers you suggested (cited references 30 and 31) and have written them as follows in Lines 86 to 91 in the main text as shown below.

“Technological advances over the past decade have led to the development of a variety of new devices in the head and neck region, including the EXOSCOPE, to achieve a less invasive, more accurate, and safer approach for preserving function after treatment [30]. However, the gold standards for assessing swallowing function remain videoendoscopy (VE) and videofluoroscopy (VF) [31, 32], and the methods described above for assessing breathing and swallowing coordination are impractical.”

5. Discuss prior research that has analyzed breathing-swallowing patterns in patients with dysphagia using noninvasive monitoring methods.

Response: Thank you very much for your very valuable opinion. Important information was missing. According to this comment, we have written the following on lines 91-99.

“Recently, Yagi et al. developed a swallowing monitoring system as a non-invasive method to examine swallowing-respiration coordination [33]. This system has made it possible to systematically evaluate swallowing sounds, laryngeal movements during swallowing, and coordination of swallowing and breathing at the bedside. Subsequent studies using this system revealed that patients with dysphagia tend to have prolonged swallowing latency and pause duration, and exhibit I-SW or SW-I patterns that reflect dyscoordination of breathing and swallowing [29]. Furthermore, previous studies have shown that breathing-swallowing discoordination is a strong independent predictor of exacerbations in patients with chronic obstructive pulmonary disease (COPD), suggesting that breathing-swallowing discoordination warrants early detection and intervention [34].”

Methods

6. Explain swallowing evaluation methods like Food Intake Level Scale, Hyodo score, Penetration-Aspiration Scale in more detail.

Response: There was insufficient explanation for each item. According to this comment, we have added details.

7. Provide more details on the swallow monitoring system - sensors used, data acquisition, analysis software etc.

Response: Thank you very much for your very valuable opinion. There was insufficient explanation for each item. According to this comment, we have added details on lines 137-148 as shown below. 

”Three signal components were recorded by a swallowing monitoring system to detect and evaluate swallowing activity [33]. Figure 1 shows the analytical instrument that was used, and its attachment. Laryngeal motion and swallowing sounds were simultaneously recorded using a custom-made piezoelectric sensor attached to the thyroid cartilage. The sensor is a piezoelectric film (detector size: 10 × 30 mm) that generates an electric charge when bent, and has a wide dynamic range between 0–4 kHz. Respiratory flow was measured using a nasal cannula-type flow sensor (Pro-Tech ProFlow cannula, Sleep Lab Products, USA) and differential pressure transmitter (KL-17; Nagano Keiki, Japan), and was recorded at 1 kHz. Laryngeal motion was recorded at 1 kHz, and the sound signal was recorded at 10 kHz and stored simultaneously with the respiratory signal in a Micro SD card for later analysis. Additionally, we recorded the timing of swallowing for later verification using a foot switch to generate TTL-level pulse signals. The signals were analyzed using MATLAB (R2014b, MathWorks, USA) on a 64-bit Windows 8 professional computer.“

8. Clearly define all swallowing parameters that were quantified - latency, laryngeal motion, respiratory phases etc.

Response: Thank you very much for your very valuable opinion. There was insufficient explanation for each word. According to this comment, we have added details on lines 152-172 as shown below. 

“The relationship between swallowing and respiratory cycles was assessed using parameters, including swallowing latency, laryngeal rising time (LRT), laryngeal activation duration (LAD), pause duration, old phase, and co-phase, as previously described [27, 29, 33, 39]. 

The parameters were defined as follows:

Swallowing latency: The time from the onset of respiratory pause to the onset of the swallowing reflex, defined as the time point when the laryngeal elevation speed reaches its maximum. 

LRT: The time required for the larynx to elevate to its highest position. 

LAD: The duration between the time of maximal laryngeal elevation speed and return of the larynx to its initial position. 

Pause duration: The duration of respiratory pause associated with swallowing. 

Old phase: Timing of swallowing in the respiratory cycle, expressed as the onset of the preceding inspiration to the maximum speed of laryngeal elevation (swallowing onset), normalized by the mean length of the respiratory cycle being 1. 

Co-phase: The time from the onset of swallowing to immediately following inspiration, normalized by the mean length of the respiratory cycle being 1.

Breathing–swallowing (B-SW) coordination pattern: We classified the breathing–swallowing pattern based on two sets of parameters: (1) B-SW type, which characterizes the interplay between swallowing and the preceding respiratory phase, either E-SW (expiration-swallow) or I-SW (inspiration-swallow); and (2) SW-B type, which characterizes the relationship between swallowing and subsequent respiratory phase, either SW-E (swallow-expiration) or SW-I (swallow-inspiration).

”

9. Describe the food and liquid stimuli used for swallow trials along with preparation protocols.

Response: According to this comment, we have added details on lines 172-175 as shown below.

“Test foods: We utilized soft jelly and water as test foods, ensuring that the soft jelly properties, such as hardness, adhesiveness, and cohesiveness, strictly adhered to the criteria outlined in the Japanese Society of Dysphagia Rehabilitation specification for level 0 dysphagia diet in the International Dysphagia Diet Standardization Initiative report [40]” 

10. Elaborate on participant instructions prior to swallow trials and precautions taken.

Response: According to this comment, we have written the following in lines 174 and 179 of the main text.

“Participants were positioned upright in a chair, and they voluntarily swallowed approximately 3 g of level 0 test food from a teaspoon and 3 mL of water from a 5-mL syringe, repeating the process two to five times each. The participants were instructed to swallow the level 0 jelly without chewing. During the examination, the patients were not provided with specific verbal instructions regarding the nature or timing of their swallowing or breathing patterns relative to swallowing.”

11. Explain the statistical tests used for comparing swallowing metrics before and after CRT.

Response: According to this comment, we have described the tests used, including the method for calculating the effect size in lines 182 and 192 of the main text as shown below.

 “To compare variables before and after the CRT, t-tests were performed for swallowing latency, LRT, LAD, pause duration, old phase, and co-phase. Wilcoxon rank-sum tests were performed for the FILS, VE, VF, and EAT10. Comparisons of swallowing patterns (SW-E or SW-I) were performed using χ2 or Fisher’s exact tests. The effect size (Cohen’s d) was calculated to assess the substantive impact of CRT on each variable [41]. The categorization of effect sizes, as measured by Cohen’s d, designates a lack of effect when <0.2, a small effect ranging between 0.2–0.5, a moderate effect spanning between 0.5–0.8, and a substantial effect at ≥0.8. The effect size for swallowing patterns was calculated using φ coefficient, which indicates correlation [42]. The categorization of effect sizes, as measured by φ coefficient, designates a lack of effect when <0.1, a small effect ranging between 0.1–0.3, a moderate effect spanning between 0.3–0.5, and a substantial effect at ≥0.5. All statistical analyses were performed using JMP11 software (SAS Institute Inc., Cary, NC, USA). In all instances, a two-sided p-value of <0.05 was considered statistically significant. 

” 

Results

12. When reporting changes in swallowing metrics before and after CRT, include exact p-values and effect sizes for the comparisons.

Response: According to this comment, effect sizes were calculated and reported in the table and text.

13. For swallowing coordination patterns, provide percentages in addition to p-values when comparing groups.

Response: According to this comment, percentages and P values are listed. Thank you very much for making it very easy to understand. we have written the following in lines 227 and 230 of the main text. 

“After the CRT, significantly more SW-I patterns were observed in level 0 food, water, and all conditions than before CRT, with statistical significance of p = 0.0025 (0% vs. 9.2%), p = 0.019 (1.01% vs. 8.57%), and p = 0.0001 (0.52% vs. 8.85%), respectively.”

14. Consider including a table comparing swallowing parameters and coordination patterns before and after CRT.

Response: According to this comment, we combined tables 3 and 4. Visibility has improved, thank you for pointing it out.

15. Mention any differences between liquid and solid swallows in the effects of CRT on breathing-swallowing coordination.

Response: Thank you for your advice. we have listed it in the discussion section.

Discussion

16. Interpret the prolonged swallowing latency and pause duration after CRT in context of known pathophysiological effects of chemoradiation on pharyngeal muscles and sensation.

Response: 

According to this comment, we have included the following in the discussion section in lines 248 and 251 as shown below. 

“Our results showed significantly prolonged swallowing latency and pause duration after CRT compared with those before CRT. This is due to the previously reported side effects of CRT, such as decreased laryngeal perception, xerostomia, radiation-induced fibrosis of the mucosa associated with the pharyngeal wall, and decreased motility of the oropharynx [9, 10, 15, 16].”

17. Compare the magnitude of increase in SW-I pattern

---

## [Decision Letter · Decision Letter 1]

15 Feb 2024

PONE-D-23-27897R1Breathing-swallowing discoordination after definitive chemoradiotherapy for head and neck cancers is associated with aspiration pneumoniaPLOS ONE

Dear Dr. Yoshida,

Thank you for submitting your manuscript to PLOS ONE. After careful consideration, we feel that it has merit but does not fully meet PLOS ONE’s publication criteria as it currently stands. Therefore, we invite you to submit a revised version of the manuscript that addresses the points raised during the review process.

As you can see there are still several points that the reviewers would like to see clarified. The recommendation to include an addition figure is one that I leave to the discretion of the authors.

We look forward to receiving your revised manuscript.

Kind regards,

Randall J. Kimple

Academic Editor

PLOS ONE

Journal Requirements:

Reviewers' comments:

Reviewer's Responses to Questions

**Comments to the Author**

1. If the authors have adequately addressed your comments raised in a previous round of review and you feel that this manuscript is now acceptable for publication, you may indicate that here to bypass the “Comments to the Author” section, enter your conflict of interest statement in the “Confidential to Editor” section, and submit your "Accept" recommendation.

Reviewer #1: (No Response)

Reviewer #3: (No Response)

2. Is the manuscript technically sound, and do the data support the conclusions?

Reviewer #1: (No Response)

Reviewer #3: Partly

3. Has the statistical analysis been performed appropriately and rigorously? 

Reviewer #1: (No Response)

Reviewer #3: (No Response)

4. Have the authors made all data underlying the findings in their manuscript fully available?

Reviewer #1: (No Response)

Reviewer #3: No

5. Is the manuscript presented in an intelligible fashion and written in standard English?

Reviewer #1: (No Response)

Reviewer #3: Yes

6. Review Comments to the Author

Reviewer #1: The authors successfully addressed all the comments provided. I believe the manuscript can be accepted in its present form.

Reviewer #3: Thank you for the opportunity to review this novel work on “Breathing-Swallowing Discoordination After Definitive Chemoradiotherapy for Head and Neck Cancers is Associated with Aspiration Pneumonia”. This is an important topic given the severity and consequences of dysphagia in persons with head and neck cancer and I commend the authors for undertaking this work. I have some questions and comments about the methodology and presentation of findings that should be addressed prior to publication.

Introduction

• [Line 85] Authors should define abbreviations for I-SW and SW-I here as they have not been previously defined.

• [Line 88] Please clarify in this sentence that the technological advances referred to, specifically the exoscope, are surgical options.

• [Line 78] In this section, authors may wish to discuss prior work by Hutcheson and colleagues re: cough flow production and respiratory muscle strength in head and neck cancer patients.

• Authors should highlight, at least in the introduction, that this manuscript refers to oropharyngeal dysphagia (rather than esophageal dysphagia).

Methods and Materials

• Patients and Methods

o Please describe if inclusionary/exclusionary criteria included prior medical history or current diagnosis of dysphagia and/or respiratory disease that may increase risk for baseline respiratory-swallow discoordination.

• Swallowing Evaluation

o [Line 120] Please describe how the Food Intake Level Scale rating was obtained in this study (i.e., via clinical interview, chart review, etc.)

o [Line 125] Please also describe how the Hyodo score and Penetration-Aspiration Scale scores were assessed and/or scored – did the participants complete an instrumental evaluation of swallowing during which the larynx was directly visualized (i.e., fiberoptic endoscopic evaluation of swallowing or a videofluoroscopic swallow study)? It is unclear from the current text whether direct visualization of the larynx was completed during this study. Given that each of these measures were validated for endoscopic examination of swallowing or videofluoroscopic swallow study analysis, use of these outcomes without direct visualization may lead to incorrect conclusions. This should be stated for transparency to the reading audience.

In addition, the authors should further clarify whether the Penetration-Aspiration Score used as part of analysis was derived as the worst score across all trials of a single consistency, the median score, or something else.

o [Line 132] The authors should further expand upon the description of the Eating Assessment Tool, including what specifically was included in statistical analysis (i.e., component scores vs total score).

o Were any measures associated with swallowing efficiency collected (i.e., swallowing residue)? Given that there is an increased risk for aspiration of post-swallow pharyngeal residue in this patient population, this may be important to mention.

o Given the timeframe post-CRT completion, were there any measures associated with CRT that were captured (i.e., pain or discomfort)?

• Monitoring of Swallowing

o [Line 172] If a specific type or brand was used for the IDDSI 0 test food, please describe this in greater detail.

o [Line 177] Please clarify the reason for the variability in the administered protocol for the test foods. This is currently described as “two to five times each” which appears to represent a large range of boluses administered. If the reason in variability is due to stopping criteria associated with patient safety, it would be helpful to have that described here.

o If space permits, this manuscript may benefit from a diagram depicting measures that were taken at different timepoints and associated assessments for better clarification.

Statistical Methods

• Mean EAT-10 total scores have traditionally been used as part of pre-post analyses similar to this. Clarification regarding whether the total score was used for analysis would better assist with identifying appropriateness of the statistical methods.

Results

• Swallowing Evaluation

o This section would benefit from further interpretation regarding the change observed following CRT. Please provide further explanation regarding findings for each of the outcome measures.

o Table 2

It appears that this table depicts “VF” as representing the Penetration-Aspiration Scale, which is scored in integer values from 1 to 8. However the range of scores in the line describing VF suggests scoring starting at 0; please clarify if these values are listed as intended.

o Table 3

Percentage symbols are missing in the Parameters column for Water and All conditions.

Punctuation is missing in several columns of this table, specific to the score ranges, such as “)”

• [Line 241] Please describe how diagnosis of aspiration pneumonia was confirmed (i.e., chart review, patient report, etc.)

Discussion

• [Line 254, 282] Although laryngeal perception, xerostomia, etc., are known pathophysiologic effects of chemoradiation on pharyngeal muscles, these items were not reported as measured during this specific study. Based on what is included currently, the data do not appear to support the causes of the change in swallowing latency and pause duration. The authors may wish to review the phrases describing these relationships.

• [Line 301] Please elaborate on limitations of this study, including its observational design and lack of a control group for comparison. If there were discrepancies between the bolus protocols across study timepoints (i.e., a difference in the total number of swallow trials), how might that effect the percentages observed in Table 3?

• How do findings from this work compare/align to those of Hopkins-Rossabi and colleagues (Respiratory-swallow coordination and swallowing impairment in head and neck cancer, 2021)?

7. PLOS authors have the option to publish the peer review history of their article (what does this mean?). If published, this will include your full peer review and any attached files.

Reviewer #1: No

Reviewer #3: No

---

## [Author Response · Author response to Decision Letter 1]

12 Mar 2024

Dear Editors and Reviewers

We sincerely appreciate the editor’s and reviewers’ comments. We revised our article according to the comments. 

Reviewer #3: 

Response: Thank you for your thoughtful review. 

Comment:

1. [Line 85] Authors should define abbreviations for I-SW and SW-I here as they have not been previously defined.

Response: Thank you for your valuable opinion. Corrections have been made.

2. [Line 88] Please clarify in this sentence that the technological advances referred to, specifically the exoscope, are surgical options.

Response: According to this comment, Corrections have been made. [Line 90-94]

3. [Line 78] In this section, authors may wish to discuss prior work by Hutcheson and colleagues re: cough flow production and respiratory muscle strength in head and neck cancer patients.

Response: According to this comment, we have revised the following with citations as follows.

In previous studies, it has been reported that patients with dysphagia following radiotherapy for head and neck cancer exhibited decreased cough strength and expiratory force, highlighting the significance of expiration as a crucial defensive factor in swallowing. [Line 85-88]

Hutcheson KA, Barrow MP, Warneke CL, Wang Y, Eapen G, Lai SY, et al. Cough strength and expiratory force in aspirating and nonaspirating postradiation head and neck cancer survivors. Laryngoscope. 2018;128(7):1615-21. Epub 2017/11/09. doi: 10.1002/lary.26986. PubMed PMID: 29114887; PubMed Central PMCID: PMCPMC5940582.

In line 78, the phrase "Swallowing in the Oral and Pharyngeal Phases Prior to the Esophageal Phase" was added.

4. Please describe if inclusionary/exclusionary criteria included prior medical history or current diagnosis of dysphagia and/or respiratory disease that may increase risk for baseline respiratory-swallow discoordination.

Response: Thank you for your valuable comments. 

We have replaced the term "respiratory failure" with the term "respiratory disease" in the wording of the exclusion criteria and added that the patient must not have swallowing dysfunction prior to treatment. Reasons for exclusion were also added.

Patients with previous medical history or current diagnosis of dysphagia, respiratory disease, multiple concurrent primary cancers, chronic heart failure, uncontrolled infections, or autoimmune diseases were excluded from the study due to the potential augmentation of the risk for baseline breathing–swallowing discoordination. [Line 115-118]

5. [Line 120] Please describe how the Food Intake Level Scale rating was obtained in this study (i.e., via clinical interview, chart review, etc.)

Response: Thank you for your valuable comments. According to this comment, in line 130, the phrase " The food intake level scale rating was obtained in via clinical interview and chart review" was added.

6. [Line 125] Please also describe how the Hyodo score and Penetration-Aspiration Scale scores were assessed and/or scored – did the participants complete an instrumental evaluation of swallowing during which the larynx was directly visualized (i.e., fiberoptic endoscopic evaluation of swallowing or a videofluoroscopic swallow study)? It is unclear from the current text whether direct visualization of the larynx was completed during this study. Given that each of these measures were validated for endoscopic examination of swallowing or videofluoroscopic swallow study analysis, use of these outcomes without direct visualization may lead to incorrect conclusions. This should be stated for transparency to the reading audience.

In addition, the authors should further clarify whether the Penetration-Aspiration Score used as part of analysis was derived as the worst score across all trials of a single consistency, the median score, or something else.

Response: I would like to thank you for your very valuable comments. We incorrectly stated that the Hyodo Score is based on direct observation and evaluation of the larynx by endoscopy. We have corrected it as follows. In addition, the evaluation and scoring methods for the Hyodo Score and Penetration-Aspiration Score are described. Thank you very much for your comment. [Line 133-149]

The Hyodo score was obtained based on endoscopic evaluation of swallowing with direct visualization of the larynx. This method comprises four parameters: (1) the accumulation of saliva in the vallecula and piriform sinuses, (2) the induction of the glottal closure reflex by stimulating the epiglottis or arytenoid with an endoscope, (3) the initiation of the swallowing reflex measured by the timing of "white-out" (defined as the period during which the endoscopic image is obscured due to pharyngeal closure), and (4) the clearance of the pharynx after swallowing blue-dyed water. Each parameter is evaluated on a 4-point scale (0; normal, 1; mildly impaired, 2; moderately impaired, 3; severely impaired). The Hyodo score is the sum of scores for these parameters, ranging from 0 to 12. Patients with a score below 5 are considered to have normal swallowing function and can consume food orally without restrictions. Patients with a score above 8 are deemed to have severe swallowing dysfunction and are not permitted any oral intake [37]. 

Penetration-aspiration scale (PAS): PAS is an eight-point scale used to characterize both the location of airway invasion events and a patient’s response with 1 representing the least and 8 representing the highest or most severe score. PAS scores encompass several observations within each score assessed by VF : (1) the depth of airway invasion (material positioned above, in contact with, or below the level of the vocal folds); (2) the presence or absence of material remaining after the swallow (ejected or not ejected); and (3) the patient’s response to material in the airway (efforts to clear the material) [38]. In this study, the PAS were based on the worst values observed during two periods: pre-CRT and post-CRT.

7. [Line 132] The authors should further expand upon the description of the Eating Assessment Tool, including what specifically was included in statistical analysis (i.e., component scores vs total score).

Response: Thank you for your valuable comments. The total score was evaluated in the statistical analysis. According to this comment, we have corrected it as follows.

The EAT-10 questionnaire comprises 10 inquiries, as delineated below, with responses graded on a scale from 0 to 4, and the total score was evaluated in the statistical analysis: (1) My swallowing problem has caused me to lose weight; (2) My swallowing problem interferes with my ability to go out for meals; (3) Swallowing liquids takes extra effort; (4) Swallowing solids takes extra effort; (5) Swallowing pills takes extra effort; (6) Swallowing is painful; (7)The pleasure of eating is affected by my swallowing; (8) When I swallow food sticks in my throat; (9) I cough when I eat; and (10) Swallowing is stressful. The cumulative score ranges between 0 and 40. Elevated scores correlate with heightened severity of dysphagia, whereas diminished scores denote a milder manifestation of dysphagia. A score of ≥3 is considered abnormal. 

[Line 152-161]

8. Were any measures associated with swallowing efficiency collected (i.e., swallowing residue)? Given that there is an increased risk for aspiration of post-swallow pharyngeal residue in this patient population, this may be important to mention.

Response: Thank you for your valuable comments. We wrote it as limitation in lines as 346 to 349 of the main text as shown below. 

“In this study, no assessment was conducted on indicators related to swallowing efficiency, such as swallowing residue. For patients at higher risk of aspirating post-swallow pharyngeal residue, such as in this study, it was deemed necessary to include the evaluation of swallowing residue as an additional assessment item for a more comprehensive evaluation.“

9. Given the timeframe post-CRT completion, were there any measures associated with CRT that were captured (i.e., pain or discomfort)?

Response: Thank you for your valuable comments. We wrote it as limitation in lines as 349 to 351 of the main text as shown below. 

The measurements related to CRT such as pain and discomfort were not incorporated as evaluation criteria in accordance with the protocol. The possibility that these items could affect the results of this study cannot be ruled out.

10. [Line 172] If a specific type or brand was used for the IDDSI 0 test food, please describe this in greater detail.

Response: Thank you for your valuable comments. According to this comment, we wrote it in lines as 197 to 200 of the main text as shown below. 

We utilized soft jelly called ENGELEADR-grape (Otsuka Pharmaceutical Factory Inc.) and water as test foods, ensuring that the soft jelly properties, such as hardness, adhesiveness, and cohesiveness, strictly adhered to the criteria outlined in the Japanese Society of Dysphagia Rehabilitation specification for level 0 dysphagia diet in the International Dysphagia Diet Standardization Initiative report [40].

11. [Line 177] Please clarify the reason for the variability in the administered protocol for the test foods. This is currently described as “two to five times each” which appears to represent a large range of boluses administered. If the reason in variability is due to stopping criteria associated with patient safety, it would be helpful to have that described here.

Response: Thank you for your valuable comments. According to this comment, we wrote it in lines as 203 to 205 of the main text as shown below. 

The reason for some variation in the protocol for administering the study diet is the establishment of criteria for halting the test in cases where clear signs of dysphagia are present and patient safety is deemed compromised.

12. If space permits, this manuscript may benefit from a diagram depicting measures that were taken at different timepoints and associated assessments for better clarification.

Response: Thank you very much for your very constructive comments. We have carefully and positively considered the issue, but due to space limitations, we have refrained from making any changes based on the comments received. We apologize for not being able to respond to your comments.

13. Mean EAT-10 total scores have traditionally been used as part of pre-post analyses similar to this. Clarification regarding whether the total score was used for analysis would better assist with identifying appropriateness of the statistical methods.

Response: According to this comment, we have indicated that the total score was used in the analysis.

14. This section would benefit from further interpretation regarding the change observed following CRT. Please provide further explanation regarding findings for each of the outcome measures.

Response: Thank you for your valuable comments. According to this comment, we wrote it in lines as 239 to 244 of the main text as shown below.

The FILS decreased, indicating that patients encountered some trouble with ingestion after CRT (p=0.0009). Additionally, there was a significant increase in the Hyodo Score, suggesting impaired safe swallowing coordination as an objective finding on endoscopic examination after CRT (p=0.0454). There was also a significant worsening observed in EAT10 after treatment, indicating a trend where patients themselves had trouble with swallowing after CRT (p=0.0304). However, no significant changes were observed in the PAS (VF)

15. Table 2 

It appears that this table depicts “VF” as representing the Penetration-Aspiration Scale, which is scored in integer values from 1 to 8. However, the range of scores in the line describing VF suggests scoring starting at 0; please clarify if these values are listed as intended.

Response: As you pointed out, it was a typo. This has been corrected. The notation of PAS has also been added. Thank you for your comment.

16. Table 3

・Percentage symbols are missing in the Parameters column for Water and All conditions.

・Punctuation is missing in several columns of this table, specific to the score ranges, such as “)”

Response: As you pointed out, there were omissions and mistakes. We have corrected it. Thank you for your comments.

17. [Line 241] Please describe how diagnosis of aspiration pneumonia was confirmed (i.e., chart review, patient report, etc.)

Response: According to this comment, Corrections have been made. 

[Line 274-276] The diagnosis of aspiration pneumonia was made comprehensively, considering the patient's symptoms, clinical course including swallowing function leading to pneumonia, findings of pneumonia and its location on Computed Tomography imaging, and blood tests.

18. [Line 254, 282] Although laryngeal perception, xerostomia, etc., are known pathophysiologic effects of chemoradiation on pharyngeal muscles, these items were not reported as measured during this specific study. Based on what is included currently, the data do not appear to support the causes of the change in swallowing latency and pause duration. The authors may wish to review the phrases describing these relationships.

Response: Thank you for your valuable comments. We believe that the decline in laryngeal perception is evaluated, even though not directly, by assessing parameters such as cough reflex , swallowing initiation, and saliva retention, in other words, the total score of the HYODO score. Additionally, quantitative measurement of saliva secretion, as you pointed out, was not included in this study's observations. Based on the above, as shown below, we have corrected the categorical expressions.

[Line 288-290] This could potentially be attributed to reported adverse effects of CRT, such as decreased laryngeal perception, xerostomia, radiation-induced fibrosis of the mucosa associated with the pharyngeal wall, and decreased motility of the oropharynx [9, 10, 15, 16].

19. [Line 301] Please elaborate on limitations of this study, including its observational design and lack of a control group for comparison. If there were discrepancies between the bolus protocols across study timepoints (i.e., a difference in the total number of swallow trials), how might that effect the percentages observed in Table 3?

Response: Thank you for your valuable comments. According to this comment, we added and revised the Limitation as shown below.

Based on the observational design of this study, there is a potential for selection bias, information bias, and involvement of latent confounding factors. Furthermore, a limitation arises from the absence of comparable control groups for comparison. [Line 342-344]

Additionally, for patient safety, the protocol states that the number of swallows was 2-5 times, but in the case group that actually went into the analysis, there were no differences in the bolus protocol between the time points of the study.For this reason, we believe that the data in Table 3 are consistent results.

20. How do findings from this work compare/align to those of Hopkins-Rossabi and colleagues (Respiratory-swallow coordination and swallowing impairment in head and neck cancer, 2021)?

Response: Thank you for providing specific references. We are confident that the discussion section of the main text will be enriched with your help. We have added the following.

Hopkins-Rossabi et al. reported that in recent years, patients with HNSCCs (head and neck squamous cell carcinomas) with dysphagia experience increased occurrences of dysphagia and penetration/aspiration as the proportion of optimal respiratory-swallow phase patterns decreases. They also noted changes in respiratory-swallow phase patterns before and after cancer treatment. Our findings in this study are consistent with the results of this previous research. [Line 346-351]

---

## [Decision Letter · Decision Letter 2]

4 Apr 2024

PONE-D-23-27897R2Breathing-swallowing discoordination after definitive chemoradiotherapy for head and neck cancers is associated with aspiration pneumoniaPLOS ONE

Dear Dr. Yoshida,

Thank you for submitting your manuscript to PLOS ONE. After careful consideration, we feel that it has merit but does not fully meet PLOS ONE’s publication criteria as it currently stands. Therefore, we invite you to submit a revised version of the manuscript that addresses the points raised during the review process.

We look forward to receiving your revised manuscript.

Kind regards,

Randall J. Kimple

Academic Editor

PLOS ONE

Journal Requirements:

Additional Editor Comments:

Please address comments of the reviewer which will help to improve the quality of the manuscript.

Reviewers' comments:

Reviewer's Responses to Questions

**Comments to the Author**

1. If the authors have adequately addressed your comments raised in a previous round of review and you feel that this manuscript is now acceptable for publication, you may indicate that here to bypass the “Comments to the Author” section, enter your conflict of interest statement in the “Confidential to Editor” section, and submit your "Accept" recommendation.

Reviewer #3: (No Response)

2. Is the manuscript technically sound, and do the data support the conclusions?

Reviewer #3: Partly

3. Has the statistical analysis been performed appropriately and rigorously? 

Reviewer #3: (No Response)

4. Have the authors made all data underlying the findings in their manuscript fully available?

Reviewer #3: (No Response)

5. Is the manuscript presented in an intelligible fashion and written in standard English?

Reviewer #3: (No Response)

6. Review Comments to the Author

Reviewer #3: Thank you for the opportunity to review this article, “Breathing-Swallowing Discoordination After Definitive Chemoradiotherapy for Head and Neck Cancers is Associated with Aspiration Pneumonia”. This is an important topic given the severity and consequences of dysphagia in persons treated for head and neck cancer. I have major concerns, questions, and comments about the methodology and presentation of findings that would need to be addressed before considering publication. Some of the comments pertain to reproducibility and improving transparency in reporting based on the FRONTIERS Framework for dysphagia reporting (https://www.frontiersframework.com/app/).

Methods and Materials

- Swallowing evaluation: I believe elaboration on how swallowing evaluations were conducted would strengthen this manuscript since it the procedures are still unclear.

- [Line 124 - 125] Videoendoscopy and Videofluoroscopy: The methods and results suggest that both a videoendoscopy and videofluoroscopy were completed by participants but it is unclear in what order these assessments were completed. Please clarify the timing of the videoendoscopic and videofluoroscopic evaluations (with respect to each other) and whether swallow monitoring was conducted concurrently.

- Study Protocols:

- Elaborate on the bolus administration protocols for the videoendoscopic and videofluoroscopic evaluations, especially if they are different from the swallow monitoring bolus protocol.

- If barium contrast was used, provide details as to the manufacturer and viscosity.

- Clarify the visualization field and positioning of participant for each of the evaluations.

- For any analyses that were completed, please elaborate on any methods for blinding of studies, raters, inter/intra-rater reliability and any training that the raters may have completed prior to study analysis.

- [Line 133 - 142] Hyodo score:

- Please specify the equipment used for endoscopic evaluation of swallowing, whether the evaluation was recorded for later review (and any equipment used for recording). If the evaluation was recorded and analyzed, please elaborate on the timing of analysis relative to the endoscopic evaluation (i.e., real-time vs post-hoc), and what software was used for analysis. It may be beneficial to review the FRONTIERS framework sections on Videofluoroscopy or Fiberoptic Endoscopic Evaluation of Swallowing for guidance.

- Clarify whether lubricant and/or anesthetic was used with any of the participants during the videoendoscopic evaluation.

- [Line 143 - 149] Penetration-Aspiration Scale

- Similar to the Hyodo score comments, this section would be strengthened with support of information on how the procedure was conducted and any analysis completed.

- [Line 151 - 160] EAT-10

- Clarify the scoring parameters on the EAT-10 (what 0 vs 4 signify).

- [Line 161] Swallowing monitoring:

- Elaborate on any synchronization of the swallow evaluation procedures and any processes for aligning events.

- Provide details on the bolus protocol that was planned for participants. Clarify the proportion of participants who were able to complete the full protocol vs those who could not.

Statistical Analyses

- If not all participants could complete the full bolus protocol for any of the swallow evaluations or swallow monitoring, how was missing data handled?

Results

- Patient characteristics:

- Table 1: What proportion of participants may have undergone surgical intervention prior to CRT?

Discussion

- The majority of the discussion appears focused on respiratory-swallow coordination; the authors should consider discussion of the other measures assessed in this study (i.e., FILS, Hyodo, PAS, EAT-10) in relation to the overall findings.

- [Line 314] I believe the following statement, if pertaining to the current study, may require further statistical support that I am unable to identify in the current draft of this manuscript: "In the present study, prolonged swallowing latency after CRT was attributed to edema, fibrous scarring, muscle weakness, and decreased laryngeal sensation due to chemoradiation treatment." If this statement is associated with this manuscript, detailed information regarding the relationship between edema, presence of fibrosis, muscle weakness, and laryngeal sensation should be provided.

- [Line 323] The authors may wish to verify Dr. Martin-Harris's name within the discussion section

- In the present study, the reported bolus protocol (line 201) is limited to two bolus viscosities of approximately the same size. What influence do the authors think having a more expansive bolus protocol may have on their findings? How does this protocol contrast to other published protocols in terms of external validity?

7. PLOS authors have the option to publish the peer review history of their article (what does this mean?). If published, this will include your full peer review and any attached files.

Reviewer #3: No

---

## [Author Response · Author response to Decision Letter 2]

16 Apr 2024

Dear Editors and Reviewers

We sincerely appreciate the editor’s and reviewers’ comments. We revised our article according to the comments. 

Thanks to the insightful feedback provided, I am confident that the manuscript has become clearer and more comprehensible. I am deeply grateful for your continued support.

Reviewer #3: 

 Response: Thank you for your thoughtful review.

Comment:

1.

Methods and Materials

- Swallowing evaluation: I believe elaboration on how swallowing evaluations were conducted would strengthen this manuscript since it the procedures are still unclear.

- [Line 124 - 125] Videoendoscopy and Videofluoroscopy: The methods and results suggest that both a videoendoscopy and videofluoroscopy were completed by participants but it is unclear in what order these assessments were completed. Please clarify the timing of the videoendoscopic and videofluoroscopic evaluations (with respect to each other) and whether swallow monitoring was conducted concurrently.

Response: Thank you for your valuable opinion. According to this comment, we have revised the following.

All swallowing evaluation were performed on the same day before and one month after CRT. [Line 125]

2..

- Study Protocols:

- Elaborate on the bolus administration protocols for the videoendoscopic and videofluoroscopic evaluations, especially if they are different from the swallow monitoring bolus protocol. 

Response: Thank you for your suggestion, I have made the corrections regarding VE and VF and their respective explanations.

For the VE evaluation, we used a nasopharyngeal-laryngoscope with a diameter of 3.9 mm with up/down tip deflection capability (Olympus ENF-VH; Olympus Tokyo, Japan) and a digital color video monitor to perform the VE. Patients in the sitting position underwent transnasal endoscopic examinations without nasal anesthetic spray. The Hyodo score was obtained based on endoscopic evaluation of swallowing using 3 ml blue-dyed water with direct visualization of the larynx. [Line 141-145]

For the VF evaluation, 3 cm3 of 40% (w/v) barium sulfate (Kaigen Pharma Co., Ltd, Osaka, Japan) with a viscosity of 16 mPa・s was injected into the participant's oral cavity while the evaluator was seated. Then, patients were observed during swallowing to determine whether there was penetration or aspiration, rated by the PAS. The visualization field of fluoroscopic examination extended from the infraorbital border to the thoracic esophagus, and evaluations were conducted twice each for anterior and lateral views. [Line 161-166]

3.

- If barium contrast was used, provide details as to the manufacturer and viscosity.

Response: Thank you for your valuable opinion. According to this comment, we have revised the following.

For the VF evaluation, 3 cm3 of 40% (w/v) barium sulfate (Kaigen Pharma Co., Ltd, Osaka, Japan) with a viscosity of 16 mPa・s was injected into the participant's oral cavity while the evaluator was seated. [Line 161]

4.

- Clarify the visualization field and positioning of participant for each of the evaluations.

Response: Thank you for your valuable opinion. According to this comment, we have revised the following.

Patients in the sitting position underwent transnasal endoscopic examinations without nasal anesthetic spray. [Line 143-144]

The visualization field of fluoroscopic examination extended from the infraorbital border to the thoracic esophagus, and evaluations were conducted twice each for anterior and lateral views. [Line 164-166]

5.

 For any analyses that were completed, please elaborate on any methods for blinding of studies, raters, inter/intra-rater reliability and any training that the raters may have completed prior to study analysis. -

Response: We appreciate your valuable comments. We have made additions and corrections according to your comments.

Three experienced Otolaryngology specialists, who have completed swallowing function evaluation training set by the Ministry of Health, Labor, and Welfare and the Society of Swallowing and Dysphagia of Japan, along with a dysphagia-certified nurse and a speech-language pathologist, assessed the swallowing function using VE and VF. All evaluations of VE and VF were recorded in audio-video interleave (AVI) files at a rate of 30 frames per second. All evaluations were discussed on the same day following the examination to achieve inter-rater agreement. VE and VF were performed safely on all patients with no apparent complications. [Line 134-140]

6.

- [Line 133 - 142] Hyodo score:

- Please specify the equipment used for endoscopic evaluation of swallowing, whether the evaluation was recorded for later review (and any equipment used for recording). If the evaluation was recorded and analyzed, please elaborate on the timing of analysis relative to the endoscopic evaluation (i.e., real-time vs post-hoc), and what software was used for analysis. It may be beneficial to review the FRONTIERS framework sections on Videofluoroscopy or Fiberoptic Endoscopic Evaluation of Swallowing for guidance.

Response: We appreciate your valuable comments. We have made additions and corrections according to your comments.

All evaluations of VE and VF were recorded in audio-video interleave (AVI) files at a rate of 30 frames per second. All evaluations were discussed on the same day following the examination to achieve inter-rater agreement. VE and VF were performed safely on all patients with no apparent complications. [Line 138-140]

7.

- Clarify whether lubricant and/or anesthetic was used with any of the participants during the videoendoscopic evaluation.

Response: We appreciate your valuable comments. We have made additions and corrections according to your comments.

patients in the sitting position underwent transnasal endoscopic examinations without nasal anesthetic spray. [Line 144]

8.

-- [Line 143 - 149] Penetration-Aspiration Scale Similar to the Hyodo score comments, this section would be strengthened with support of information on how the procedure was conducted and any analysis completed. 

Response: Thank you very much for your comment. I have also included the VF evaluation method in the answer to comment 5.

Three experienced Otolaryngology specialists, who have completed swallowing function evaluation training set by the Ministry of Health, Labor, and Welfare and the Society of Swallowing and Dysphagia of Japan, along with a dysphagia-certified nurse and a speech-language pathologist, assessed the swallowing function using VE and VF. All evaluations of VE and VF were recorded in audio-video interleave (AVI) files at a rate of 30 frames per second. All evaluations were discussed on the same day following the examination to achieve inter-rater agreement. VE and VF were performed safely on all patients with no apparent complications. [Line 134-145]

9.

Clarify the scoring parameters on the EAT-10 (what 0 vs 4 signify).

Response: We appreciate your valuable comments. We have made additions and corrections according to your comments.

The score for each item on the EAT10 is 0 indicating no abnormality and 4 indicating severe disability. [Line 175]

10.

- [Line 161] Swallowing monitoring:

- Elaborate on any synchronization of the swallow evaluation procedures and any processes for aligning events.

- Provide details on the bolus protocol that was planned for participants. Clarify the proportion of participants who were able to complete the full protocol vs those who could not.

Response: Thank you for your valuable feedback. We have added notes on the evaluation procedure, consistency measures, and bolus protocol. We also noted that no participants deviated from the protocol.

The stored data were subsequently analyzed by a medical engineering research expert and a neurophysiologist specialized in swallowing and respiration. Both analysts were blinded to patient information, and the results were determined through consensus.

[Line 191-193]

Monitoring of swallowing was performed by two experienced otolaryngology specialists on the same day as the VE and VF evaluations. Before the examination, participants were told that the test would be performed with level 0 food, followed by water, that they would be asked to swallow five times each, that they could stop if they felt pain, that they should not talk during the test, and that they should not chew level 0 food. [Line 217-221]

Monitoring of swallowing was performed safely on all patients with no apparent complications. [Line 231]

11.

Statistical Analyses

- If not all participants could complete the full bolus protocol for any of the swallow evaluations or swallow monitoring, how was missing data handled?

Response: We appreciate your valuable comments. We have added the following information.

Monitoring of swallowing was performed safely on all patients with no apparent complications. [Line 231]

12.

Results

- Patient characteristics:

- Table 1: What proportion of participants may have undergone surgical intervention prior to CRT?

Response: We appreciate your valuable comments. We have added the following information.

All patients received CRT as initial treatment and no prior surgical intervention. [Line 252]

13.

Discussion

- The majority of the discussion appears focused on respiratory-swallow coordination; the authors should consider discussion of the other measures assessed in this study (i.e., FILS, Hyodo, PAS, EAT-10) in relation to the overall findings.

Response: Thank you very much for your constructive comments. We have revised our discussion as follows.

Before and after CRT, deterioration in FOIS and EAT10 scores was observed. This suggests that due to CRT, patients experience inhibition of oral intake, indicating the need for some form of nutritional support for the maintenance of vital functions. It also indicates a decrease in quality of life from both functional and psychological perspectives. A patient-completed swallowing screening questionnaire offers a simple and effective tool for daily clinical use, allowing healthcare providers to promptly identify swallowing issues post CRT. This could help in the timely implementation of further assessments such as VF and VE, along with appropriate rehabilitation interventions. In this protocol, the VE evaluation, or Hyodo score, significantly worsened before and after CRT, whereas no significant change was noted in the PAS scale. This suggests that when assessing swallowing after CRT, PAS may not detect changes or may be obscured, particularly with a small bolus condition. While swallowing evaluation with VE can be conducted at the bedside using only an endoscope, VF necessitates relocation to the fluoroscopy room and radiation exposure. In terms of ease of performance, VE and evaluation of breathing-swallowing coordination can be considered advantageous for evaluating swallowing function after CRT. 

 In this study, only small boluses were utilized. The rationale behind this decision was to ensure consistency with the protocols and conditions of prior studies employing noninvasive swallowing measurement system. This approach aimed to investigate variations in the coordination of breathing and swallowing across diverse patient conditions. Increasing bolus size is expected to prolong the time taken for bolus transit relative to smaller boluses, resulting in extended pause duration and potentially exacerbating in breathing-swallowing discoordination. Therefore, the observed breathing-swallowing discoordination in this study may be underestimated compared to evaluations using oral intake or protocols employing larger boluses in clinical practice. Although not included in this protocol, the utilization of various test foods in VE, VF and monitoring of swallowing could allow for the evaluation of swallowing function for each food type. Moreover, by utilizing these assessments to determine the viscosity of safe and suitable foods and rehabilitation strategies based on swallowing function, and subsequently enhancing the level of oral intake, it is conceivable that the quality of life for patients after CRT may be improved. [Line 366-393]

14.

 [Line 314] I believe the following statement, if pertaining to the current study, may require further statistical support that I am unable to identify in the current draft of this manuscript: "In the present study, prolonged swallowing latency after CRT was attributed to edema, fibrous scarring, muscle weakness, and decreased laryngeal sensation due to chemoradiation treatment." If this statement is associated with this manuscript, detailed information regarding the relationship between edema, presence of fibrosis, muscle weakness, and laryngeal sensation should be provided.

Response: Thank you very much for your constructive comments. We have revised our discussion as follows.

This could potentially be attributed to reported adverse effects of CRT, such as decreased laryngeal perception, xerostomia, radiation-induced fibrosis of the mucosa associated with the pharyngeal wall, and decreased motility of the oropharynx [9, 10, 15, 16]. However, the exact mechanism was not elucidated in this study. Investigating the relationship between swallowing-related muscle strength, motility, laryngeal sensation, saliva volume, and breathing-swallowing coordination through simultaneous measurements would be a prospective focus for future research. [Line 312-317]

Although not included in this protocol, the utilization of various test foods in VE, VF and monitoring of swallowing could allow for the evaluation of swallowing function for each food type. Moreover, by utilizing these assessments to determine the viscosity of safe and suitable foods and rehabilitation strategies based on swallowing function, and subsequently enhancing the level of oral intake, it is conceivable that the quality of life for patients after CRT may be improved. [Line 386-390]

15.

- [Line 323] The authors may wish to verify Dr. Martin-Harris's name within the discussion section

Response: Thank you for pointing this out. We have corrected it correctly.

16. 

In the present study, the reported bolus protocol (line 201) is limited to two bolus viscosities of approximately the same size. What influence do the authors think having a more expansive bolus protocol may have on their findings? How does this protocol contrast to other published protocols in terms of external validity?

Response: Thank you for your constructive feedback. according to this comment, comments received, we have added the reasons that led us to this protocol, the possibility that the results may be underestimated, and the expected changes when the bolus is increased.

 In this study, only small boluses were utilized. The rationale behind this decision was to ensure consistency with the protocols and conditions of prior studies employing noninvasive swallowing measurement system. This approach aimed to investigate variations in the coordination of breathing and swallowing across diverse patient conditions. Increasing bolus size is expected to prolong the time taken for bolus transit relative to smaller boluses, resulting in extended pause duration and potentially exacerbating in breathing-swallowing discoordination. Therefore, the observed breathing-swallowing discoordination in this study may be underestimated compared to evaluations using oral intake or protocols employing larger boluses in clinical practice.　 [Line 379-390]

---

## [Decision Letter · Decision Letter 3]

2 Jun 2024

Breathing-swallowing discoordination after definitive chemoradiotherapy for head and neck cancers is associated with aspiration pneumonia

PONE-D-23-27897R3

Dear Dr. Yoshida,

We’re pleased to inform you that your manuscript has been judged scientifically suitable for publication and will be formally accepted for publication once it meets all outstanding technical requirements.

Kind regards,

Randall J. Kimple

Academic Editor

PLOS ONE

Additional Editor Comments (optional):

Reviewers' comments:

Reviewer's Responses to Questions

**Comments to the Author**

1. If the authors have adequately addressed your comments raised in a previous round of review and you feel that this manuscript is now acceptable for publication, you may indicate that here to bypass the “Comments to the Author” section, enter your conflict of interest statement in the “Confidential to Editor” section, and submit your "Accept" recommendation.

Reviewer #3: (No Response)

2. Is the manuscript technically sound, and do the data support the conclusions?

Reviewer #3: (No Response)

3. Has the statistical analysis been performed appropriately and rigorously? 

Reviewer #3: (No Response)

4. Have the authors made all data underlying the findings in their manuscript fully available?

Reviewer #3: (No Response)

5. Is the manuscript presented in an intelligible fashion and written in standard English?

Reviewer #3: (No Response)

6. Review Comments to the Author

Reviewer #3: Thank you for addressing the previous comments. The authors may wish to address the following minor edits prior to publication:

- Lines 344 - 346 and lines 402 - 404: In lines 344-346, the authors state that prolonged latency was attributed to edema, fibrous scarring, muscle weakness and decreased sensation. However, some caution should be exercised with this statement if there are no data present in this paper to support this relationship. This statement is in contrast with lines 402-404, which indicates that data on pain and discomfort (and edema, fibrosis, weakness, and sensation) were not collected.

Line 367: "FOIS" is stated in this sentence - but I believe the authors intended to write "FILS".

7. PLOS authors have the option to publish the peer review history of their article (what does this mean?). If published, this will include your full peer review and any attached files.

Reviewer #3: No
